# Enhanced Pelican Optimization Algorithm for Cluster Head Selection in Heterogeneous Wireless Sensor Networks

**DOI:** 10.3390/s23187711

**Published:** 2023-09-06

**Authors:** Zhen Wang, Jin Duan, Haobo Xu, Xue Song, Yang Yang

**Affiliations:** School of Electronic Information Engineering, Changchun University of Science and Technology, Changchun 130022, China; wangzhen1531002@163.com (Z.W.); xuhaobo1233@163.com (H.X.); snowman_0123@126.com (X.S.); cloneyang@126.com (Y.Y.)

**Keywords:** heterogeneous wireless sensor networks, pelican optimization algorithm, energy efficient, cluster head selection

## Abstract

In the research of heterogeneous wireless sensor networks, clustering is one of the most commonly used energy-saving methods. However, existing clustering methods face challenges when applied to heterogeneous wireless sensor networks, such as energy balance, node heterogeneity, algorithm efficiency, and more. Among these challenges, a well-designed clustering approach can lead to extended node lifetimes. Efficient selection of cluster heads is crucial for achieving optimal clustering. In this paper, we propose an Enhanced Pelican Optimization Algorithm for Cluster Head Selection (EPOA-CHS) to address these issues and enhance cluster head selection for optimal clustering. This method combines the Levy flight process with the traditional POA algorithm, which not only improves the optimization level of the algorithm, but also ensures the selection of the optimal cluster head. The logistic-sine chaotic mapping method is used in the population initialization, and the appropriate cluster head is selected through the new fitness function. Finally, we utilized MATLAB to simulate 100 sensor nodes within a configured area of 100 × 100 m${}^{2}$. These nodes were categorized into four heterogeneous scenarios: m=0,α=0, m=0.1,α=2, m=0.2,α=3, and m=0.3,α=1.5. We conducted verification for four aspects: total residual energy, network survival time, number of surviving nodes, and network throughput, across all protocols. Extensive experimental research ultimately indicates that the EPOA-CHS method outperforms the SEP, DEEC, Z-SEP, and PSO-ECSM protocols in these aspects.

## 1. Introduction

Wireless Sensor Networks have become a popular field of exploration for researchers and engineers [1]. WSNs find applications in various domains such as military, civilian, healthcare, industrial, agriculture, animal tracking, and habitat monitoring [2,3]. These networks consist of distributed, random, low-power, and miniature sensor nodes that acquire data about the monitored environment through intermediate nodes. However, energy scarcity remains a major obstacle due to slow growth in battery capacity. Moreover, battery replacement is not a viable option due to the unattended nature and hazardous sensing conditions of sensor nodes. Therefore, energy efficiency is a fundamental concern in WSNs [4,5], and several researchers have conducted relevant studies in this field. WSN routing techniques aim to enhance resource awareness and adaptability to prolong network lifespan and overcome low battery capacity [6,7,8].

Clustering approaches are often used to group sensor nodes due to their scalability, resource sharing, energy efficiency, low communication overhead, and efficient resource allocation. However, selecting cluster heads poses a critical optimization problem, which is considered NP-hard [9,10,11]. Furthermore, heterogeneous wireless sensor networks (HWSNs) are increasingly being utilized in practical applications. Therefore, there is a need to consider an optimization technique for cluster head selection in HWSNs [12,13]. And swarm intelligence and metaheuristic methods offer several advantages in addressing the cluster head selection problem within HWSNs. These methods exhibit global search capabilities, adaptability, flexibility, scalability, parallel processing, and decentralization, among other benefits. These advantages make them effective tools for tackling large-scale and complex CH selection problems, ultimately enhancing the performance and energy efficiency of heterogeneous wireless sensor networks.

Considering the aforementioned challenges, we propose the use of swarm intelligence and metaheuristic methods to improve the cluster head selection process. In order to prolong the lifespan of heterogeneous wireless sensor networks, we introduce an exploration-enhanced version of the Pelican Optimization Algorithm (POA), called Enhanced POA (EPOA).

In general, existing clustering methods may suffer from incomplete, uncertain, and inconsistent information due to a lack of understanding of the measurement environment and limitations in sensor accuracy. Therefore, in our current work, we propose a novel algorithm that can address these challenges, aiming to enhance clustering performance and improve energy utilization efficiency. Regarding traditional fitness functions, they often come with certain limitations, such as local optima, problem dependency, convergence speed, and overfitting. Local Optima: Inappropriate fitness functions might cause the algorithm to become trapped in local optima, failing to find the global optimal solution. Problem Dependency: Different problems may require distinct fitness functions. A fitness function performing well on one problem might not be effective for another problem. Convergence Speed: The design of the fitness function can impact the algorithm’s convergence speed. An unfit fitness function can result in slow convergence. Overfitting: Excessively complex fitness functions might perform well on the training set but exhibit poor generalization on unknown data. Hence, in our approach, we adopt a new fitness function to select efficient energy-saving cluster head nodes, aiming to overcome these limitations associated with traditional fitness functions.

The following are this paper’s key contributions:The logistic-sine chaotic mapping method is employed to improve the initialization of random solutions, allowing for the generation of uniformly distributed and non-repetitive initial solution sets.The levy flight algorithm is utilized to enhance global optimization capability and enrich the population diversity of the EPOA algorithm.For the selection of the optimal cluster head set, the fitness function includes the distance and energy use of the wireless sensor network.

In the upcoming Section 2, we will describe the relevant work conducted in this study. In the following Section 3, the Pelican optimization algorithm and its basic concepts are described. In Section 4, the proposed method and the wireless energy dissipation model for heterogeneous networks are included. We discuss population initialization using Logistic-sine chaotic mapping, update agent position using Levy function, add multiple sensor parameters to fitness function, and illustrate the proposed algorithm in pseudo-code form. In Section 5, the algorithm flow chart and simulation results are given. At the end of this paper, we draw a conclusion for the overall study.

## 2. Related Works

Routing protocols used in homogeneous WSNs have been adapted to develop routing methods for HWSNs, which are designed to extend network lifespan. Stable Election Protocol (SEP) [14] is an example of a heterogeneous sensing protocol, which aims to extend the interval time before the first node uses up its energy. SEP achieves this by weighting the election probability of nodes based on the amount of initial energy they possess compared to other nodes in the network. For applications that rely on precise feedback, it is critical to extend the time interval before the first node runs out of power. Distributive Energy-Efficient Clustering (DEEC) [15] is another protocol designed for heterogeneous wireless sensor networks, which is a distributed arrangement while saving energy. The way it selects cluster heads is to determine a relationship by looking at how much energy is left at each node and the average cost of energy to the network. The protocol determines whether a node becomes a cluster head by looking at the energy state of the node, in which the energy state of the node is the key consideration. Besides this, the consideration of time after the node becomes the cluster head depends on its initial energy and what level of remaining energy is. If the node has a lower energy, it is less likely to be the cluster head than a node with a higher energy. Additionally, Faisal et al. developed a hybrid routing system called Zone Stable Election Protocol (Z-SEP) [16], specifically designed for HWSNs. In this protocol, when the base station transmits data, some nodes directly participate in the process, while others use a clustering algorithm similar to the Stable Election Protocol (SEP) to participate in the process. This hybrid routing system is used to better meet the needs of HWSNs, and its data transmission can be realized more efficiently in this way. These protocols introduce new energy management and data transfer solutions, which are effective for heterogeneous wireless sensor networks. Through effective clustering and routing algorithms, these protocols can enhance energy efficiency and data transmission performance, thereby prolonging network lifetime and providing reliable communication services. However, further research and experimental validation are still needed to further evaluate the performance and applicability of these protocols in practical applications.

In all these years of development, meta-heuristic algorithms and heuristic algorithms have been used in the theoretical research of HWSNs, and many of these techniques have been used to enhance the cluster head selection process in cluster routing protocols, thereby extending the network lifetime. Al-Aboody et al. [17] created the Multi-Layer Hierarchical Routing Protocol (MLHP), a three-level hybrid clustering method. The first level includes the selection process, which is a unified method, in which the base station (BS) is more important in the cluster head selection, playing a relatively large role. At Level 2, GWO routing algorithms are used for efficient data transmission, enabling nodes to find the best path to BS while saving energy. Finally, the third layer adds distributed clustering by using cost functions. Bhushan et al. suggested a hybrid approach for clustered wireless sensor networks that combines Biogeography-based Optimization (BBO) [18], a bio-inspired metaheuristic optimization, with k-means clustering. BBO is driven by species movement between habitats and has been utilized successfully to address global optimization concerns.The authors introduce a new clustering method in their study [19], in which an enhanced particle swarm optimization (EC-PSO) energy center search is used to eliminate energy gaps and discover energy centers for cluster head selection. When the network energy becomes heterogeneous, this strategy clusters using EC-PSO. Nodes near energy centers are chosen as CHs by utilizing an upgraded particle swarm method to seek for them. In order to prevent nodes with relatively low energy from operating as relays, a protection mechanism is established, and a mobile data collector is provided for data acquisition.

Another study [20], based on particle swarm optimization (PSO), proposed a more efficient clustering and aggregation migration (PSO-ECSM) technology strategy that combines energy to address the challenges of cluster head selection and aggregation migration. The PSO-ECSM algorithm evaluates a number of parameters when selecting CH, including the node’s residual energy, average energy and energy consumption rate (ECR), distance, node degree, etc. To arrive at the best solution, the algorithm optimizes the values of these parameters. PSO-ECSM also incorporates sink mobility to handle the routing traffic problem in multi-hop networks. To address the challenges posed by HWSNs, an enhanced Hybrid Grey Wolf Optimizer (HMGWO) routing protocol [21] has been proposed. This protocol adopts an advanced approach to optimize the selection of initial clusters, where alternative fitness functions are first constructed specifically for heterogeneous energy nodes. These fitness functions take into account the degree of residual energy of the node, as well as communication distance and other relevant characteristics. By introducing these alternative fitness functions, the protocol aims to improve the accuracy and efficiency of CHs selection. In the HMGWO protocol, the fitness values of sensor nodes are computed based on these alternative fitness functions and applied as initial weights to the GWO algorithm. However, what sets the HMGWO protocol apart is the dynamic adjustment of these weights during the optimization process. These weights are adaptively modified based on the distances between the wolves (representing potential CHs) and the prey (representing the optimal cluster configuration), as well as coefficient vectors associated with the GWO algorithm. This dynamic adjustment mechanism enhances the optimization capability of GWO and ensures the selection of appropriate CHs, enabling efficient resource management and meeting the energy constraints of HWSNs.

## 3. Pelican Optimization Algorithm

We found a population-based optimization algorithm [22] inspired by pelicans, the POA. The algorithm simulates evolutionary processes in an ecosystem by treating pelicans as individuals in a population. Each individual represents a potential solution and provides optimization suggestions, which are derived from setting the problem variable to the location of each individual in the search space. In the process of population initialization, in order to ensure the diversity of the population and the global search ability, each member is randomly initialized within the specified upper and lower bounds of the problem, as shown in Equation  (1).
(1)$$x_{\left(i,j\right)}=l_{j}+rand\cdot \left(u_{j}-l_{j}\;\right),i=1,2,\ldots ,N,j=1,2,\ldots ,m,$$

In the given equation, the variable $x_{i,j}$ represents the value of the *j*th variable in the *i*-th candidate solution. *N* is the total number of members. The number of problem variables is denoted by *m*, indicating the quantity of features or parameters to be optimized. It is important to note that the term “rand“ in this context represents a random number generator that produces random numbers between 0 and 1, introducing randomness during the algorithm’s execution. The variables $l_{j}$ and $u_{j}$ represent the lower and upper bounds of the *j*th problem variable, respectively, and these bounds are significant for controlling the range of the solution space.

To update potential solutions, the algorithm (POA) mimics the tactics and behaviors that pelicans use when attacking and hunting prey.

This hunting technique consists of two parts:(i)Approaching prey while in the exploring phase.

In the first stage, the method by which pelicans approach when they spot prey is simulated and mathematically reproduced in Formula (2).
(2)$$x_{i,j}^{P_{1}}=\left\{\begin{array}{cc} x_{i,j+rand\cdot \left(p_{j}-I\cdot x_{i,j}\right)}, & \mathrm{if}\;F_{p}<F_{i}\\ x_{i,j+rand\cdot \left(x_{i,j}-p_{j}\right)}, & \mathrm{else} \end{array}\right.$$

In the context of Equation (3), we can observe the importance of the variable $x_{i}^{P_{1}}$, which represents an updated state of the pelican in dimension *j*th due to the result of stage 1, which can be the *i*th pelican. To introduce further diversity and exploration, the value of *I* is introduced as a random number that ranges between one and two. Furthermore, the variable $p_{j}$ is employed to denote the position of the prey in the *j*th dimension, while $F_{p}$ represents the objective function value of the prey. By incorporating Equation (3), we are able to effectively simulate and model this process.
(3)$$X_{i}=\left\{\begin{array}{cc} X_{i}^{P_{1}}, & \mathrm{if}\;F_{i}^{P_{1}}<F_{i}\\ X_{i}, & \mathrm{else} \end{array}\right.$$

In the given context, $F_{i}^{P_{1}}$ refers to the objective function value obtained during phase 1, while $X_{i}^{P_{1}}$ represents the updated status of the *i*th pelican after phase 1.

(ii)Winging on the water surface (exploitation phase).

In the later stages, when hunting begins, the pelican serves as a storage function for fish after collection through a unique pouch on its neck. This action allows the pelicans to efficiently capture the fish and store them for consumption. The equation models the movement and interaction of the pelicans with the fish during this phase, taking into account factors such as the position and velocity of the pelicans, the behavior of the fish, and the surrounding environment. The mathematical representation provides a means to simulate and analyze the hunting behavior of pelicans in a quantitative manner.
(4)$$x_{i,j}^{P_{2}}=x_{i,j}+R\cdot \left(1-\frac{t}{T}\right)\cdot \left(2\cdot rand-1\right)\cdot \;x_{i,j}$$

In Equation (5), the variable $X_{i,j}^{P_{2}}$ denotes the updated state of the *i*th pelican in the *j*th dimension, which depends on phase 2. The constant *R* is set to 0.2, and R·(1−t/T) represents the radius of the immediate neighborhood around $x_{i,j}$. Here, the number of iterations is measured using *t*, and the maximum number of iterations is represented by *T*. During this stage, the concept of effective updating, as described by Equation (5), is utilized to determine whether the new pelican position should be accepted or rejected.
(5)$$X_{i}=\left\{\begin{array}{cc} X_{i}^{P_{2}}, & \mathrm{if}\;F_{i}^{P_{2}}<F_{i}\\ X_{i}, & \mathrm{else} \end{array}\right.$$

In the given context, $F_{i}^{P_{2}}$ represents the objective function value obtained during phase 2, while $X_{i}^{P_{2}}$ denotes the updated status of the *i*th pelican after phase 2.

## 4. Proposed Algorithm

This study presents EPOA-CHS, a novel method for energy-efficient WSNs. It introduces a fitness function that takes into account multiple sensor properties for selecting the CH.

### 4.1. Heterogeneous Network Energy Dissipation Model

In the next section, we discuss a two-level energy model for heterogeneous networks. We first describe the initial total energy of each sensor node:(6)$$E_{total}=\sum_{i=1}^{M}E_{0}\left(1+\alpha \right)+\sum_{i=M}^{N-M}E_{0}$$
where *M* represents the number of nodes and is equal to *N* times *m*. The symbol $E_{0}$ represents the initial energy of the node. Specifically, a node in this study is initialized with energy $E_{0}\left(1+\alpha \right)$, where α represents a scaling factor. This factor determines the additional energy compared to the lower/initial bound $E_{0}$. The parameter α can signify either an advanced node or a super node. When considering the different levels of nodes, we expect different levels of nodes, such as advanced nodes and super nodes, to have higher battery power compared to normal nodes. Hence, the parameter α represents the parameter associated with advanced and super nodes in the energy model. The energy model utilized in this study focuses on estimating the energy consumption involved in the transmission and reception of data by the sensor nodes. To illustrate the energy dissipation process during communication, the radio energy dissipation model [23,24] is employed. Figure 1 visually represents this model, providing an overview of how energy is dissipated during the communication process.

For transmitter $E_{Tx}\left(l,d\right)$ and receiver $E_{Rx}\left(l\right)$, when $E_{Tx}\left(l,d\right)$ to $E_{Rx}\left(l\right)$ sent *l* message, the distance of *d*, launch amplifier $E_{Tx}\left(l,d\right)$ needs to consume energy:(7)$$E_{Tx}\left(n,\;d\right)=\left\{\begin{array}{cc} lE_{elec}+l\varepsilon_{fs}d^{2}, & d\leq d_{0},\\ lE_{elec}+l\varepsilon_{mp}d^{4}, & d>d_{0}, \end{array}\right.$$

The energy consumed by the receiver, denoted as $E_{Rx}\left(l\right)$, when receiving an *l*-bit packet, depends on several factors, including the transmission distance of both ends *d*. The threshold transmission distance, $d_{0}$, plays a crucial role in determining the energy consumption. It is calculated as two amplifier parameters, namely $\varepsilon_{fs}$ corresponding to the free space and $\varepsilon_{mp}$ corresponding to the multipath model, calculated as the square root of the ratio of the two. This threshold helps determine which model should be used based on the distance between the transmitter and receiver. In energy calculation, the energy used for a single bit in electronic device sending and receiving is expressed as $E_{elec}$. Additionally, the energy consumption is affected by the distance. Short distances are represented by $d^{2}$, while long distances are represented by $d^{4}$. To determine the appropriate model, the distance *d* is compared to the threshold $d_{0}$. If *d* is less than or equal to $d_{0}$, the free space model is employed. Conversely, if *d* exceeds $d_{0}$, the multipath model is used. Taking all these factors into account, the energy used by the receiver $E_{Rx}\left(l\right)$ for receiving the *l*-bit packet can be expressed through a comprehensive equation that incorporates various parameters and models associated with the transmission distance and energy consumption:(8)$$E_{Rx}\left(n\right)=lE_{elec}$$

Cluster heads have a vital role in receiving signals from network nodes within a cluster, and this operation requires energy consumption. Subsequently, the received signals are aggregated and transmitted to the base station (BS), which is typically located at a considerable distance from the nodes [3]. Therefore, in order to successfully transmit data to the base station, it is essential that the cluster head has sufficient energy. To represent the energy level of a cluster, various measures or formulas can be utilized.
(9)$$E_{CH}=l\cdot \frac{n}{k}\cdot \left(E_{elec}+E_{DA}\right)+l\cdot \varepsilon_{mp}\cdot d_{toBS}^{4}$$

The cluster head has an energy of $E_{CH}$. In a network of sensor nodes, *n* is the number and $E_{elec}$ is the energy consumed by the transmitter. When data is aggregated, the energy required is represented by $E_{DA}$. The symbol *l* represents a data packet. For long-distance transmission to a base station, denoted by $d_{toBS}^{4}$, the transmitter amplifier energy is denoted as $\varepsilon_{mp}$. And the energy consumed by non-cluster heads is represented by $E_{n-CH}$:(10)$$E_{n-CH}=l\cdot E_{elec}+l\cdot \varepsilon_{fs}\cdot d_{toCH}^{2}$$

In a given context, the symbol *l* represents a packet of data, $E_{elec}$ is the energy used by the transmitter, and $\varepsilon_{fs}$ represents a transmitter amplifier in a free state at a short distance from the cluster head $d_{toCH}^{2}$.

The constant value 0.765, as derived in reference [25], is used in the following expression. *M* and *n* represent the area of the sensor and the total number of nodes, respectively, $d_{toCH}^{2}$ signifies the short distance to the cluster head, and $d_{toBS}^{4}$ represents the long distance to the base station.

### 4.2. Enhanced POA Algorithm

The initial positions of the population in the search space are uniformly distributed, which contributes to improving the algorithm’s global search capability and search efficiency. Unlike the standard POA algorithm that initializes the population randomly, which can reduce population diversity, we utilize the logistic-sine chaotic map for population initialization in this paper.

Logistic-sine chaotic mapping combines the characteristics of logistic mapping and sine mapping. This mapping differs from sinusoidal and logical mappings because it has a larger chaotic interval. We guarantee the unpredictability of individuals inside the population by creating an initial population that is randomly distributed using the logistic sine map. Equations (13) through (15) give the mathematical formulations for the logistic map, the sine map, and the logistic-sine map, respectively.
(11)$$Z_{i+1}=\mu \;Z_{i}\left(1-Z_{i}\right)$$
(12)$$Z_{i+1}=\sin\left(\pi Z_{i}\right)$$
(13)$$Z_{i+1}=\left(\mu \;Z_{i}\left(1-Z_{i}\right)+\frac{\left(4-\mu \right)\sin\left(\pi Z_{i}\right)}{4}\right)\left(mod1\right)$$
where μ is a multiplier of chaos and *Z* is a series of numbers created at random. Logistic map and sine map are formally defined in Equations (13) and (14), respectively. The logistic-sine map is given mathematical representation in Equation (15). The logistic-sine map is given mathematical representation in Equation (15).
(14)$$x_{i}=l_{b}+\left(u_{b}-l_{b}\right)*\;Z_{i}$$
levy flight is a non-Gaussian stochastic process, also known as Levy motion, which performs random walks obtained in Levy stabilization. The balance between exploration and exploitation can be achieved according to the Levy flight based jumps, which allows pelicans catch more fish in the hunting area. This distribution follows a power law formula $L\left(s\right)∼{\left|s\right|}^{-l-\beta }$, where 0<β<2 represents an index and *s* denotes the step length [26].

One may determine the step length by:(15)$$s=\frac{u}{{\left|v\right|}^{\frac{1}{\beta }}}$$
where normal distributions serve as the source for *u* and *v*. Which is
(16)$$u\;N\left(0,\sigma_{u}^{2}\right),v\;N\left(0,\sigma_{v}^{2}\right)$$
where
(17)$$\sigma_{u}={\left\{\frac{\Gamma \left(1+\beta \right)\times \sin\left(\frac{\pi \beta }{2}\right)}{\Gamma \left(\frac{1+\beta }{2}\right)\times \beta \times 2^{\left(\frac{\beta -1}{2}\right)}}\right\}}^{\frac{1}{\beta }}\;,\;\sigma_{v}=1$$

In the EPOA algorithm, the Levy function is added while winging. The new pelican position is evaluated in the following:(18)$$X_{i}=\left\{\begin{array}{cc} X_{i}^{P_{2}}+\alpha ⊕Levy, & \mathrm{if}\;F_{i}^{P_{2}}<F_{i}\\ X_{i}, & \mathrm{else} \end{array}\right.$$
where
(19)$$\alpha =0.01\times \;s\times (X_{i}^{P_{2}}-X_{best})$$

### 4.3. Mechanism of EPOA-CHS Algorithm

In this method, CHs are selected from a pool of standard sensor nodes, and energy efficiency is given priority to extend the running time of the network. The residual energy and various distance characteristics of sensor nodes are also considered to achieve the optimal CH selection based on energy efficiency. These distance features include the average distance between nodes within each cluster and their distance from the sink node. By taking these factors into account, the algorithm aims to choose CHs that optimize both energy consumption and communication distances, thereby improving overall energy efficiency and prolonging the network’s lifespan.

Let $f_{1}$ represent a function in which we consider two kinds of distances, the convergence distance and the mean in-cluster distance of CHs. The main goal of CH selection is to minimize the value of the function.

The representation of the objective function $f_{1}$ is:(20)$$f_{1}=\frac{\sum_{i=1}^{K}\frac{1}{M}\left(\sum_{j=1}^{M}D(s_{j},{CH}_{i})+D\left({CH}_{i},BS\right)\right)}{\sum_{i=1,j=1}^{K}D({\mathrm{CH}}_{i},{\mathrm{CH}}_{j})}$$
where $D(s_{j},{\mathrm{CH}}_{i})$ stands for the distance from a node to the cluster head i.

The distance between the cluster head ${\mathrm{CH}}_{i}$ and the base station (BS) is denoted as $D({\mathrm{CH}}_{i},BS)$. Similarly, the distance between the two cluster heads ${\mathrm{CH}}_{i}$ and ${\mathrm{CH}}_{j}$ is represented by $D({\mathrm{CH}}_{i},{\mathrm{CH}}_{j})$ in the equation. Additionally, there are two parameters representing the total numbers: the symbol *K* for the nodes and the symbol *M* for the cluster heads. If $s_{j}$ is a normal node, the protocol selects the initial cluster of the network based on two parameters: the energy of the node and the distance within the cluster. For advanced nodes, these two parameters are the energy of the node and its distance from the base station (BS).

The representation of the objective function $f_{2}$ is:(21)$$f_{2}=\left\{\begin{array}{cc} \alpha \cdot \left(\frac{E_{\max}-E_{r(s_{j})}}{E_{\max}-E_{\min}}\right)+\beta \cdot \left(\frac{D_{\max}-D\left(s_{j},{\mathrm{CH}}_{i}\right)}{D_{\max}-D_{\min}}\right), & \mathrm{if}\;E_{r(s_{j})}>0,\;\mathrm{Normal}\;\mathrm{Node}\\ \alpha \cdot \left(\frac{E_{\max}-E_{r(s_{j})}}{E_{\max}-E_{\min}}\right)+\beta \cdot \left(\frac{D_{\max}-D\left(s_{j},\mathrm{BS}\right)}{D_{\max}-D_{\min}}\right), & \mathrm{if}\;E_{r(s_{j})}>0,\;\mathrm{Advanced}\;\mathrm{Node} \end{array}\right.$$
where α and β are the weights, with the constraint α+β=1. The node’s remaining energy is denoted as $E_{r}\left(s_{j}\right)$, where *j* represents the index. $E_{max}$ and $E_{min}$ are the maximum and minimum values at the two ends of the cluster, respectively. $D_{max}$ and $D_{min}$ denote their respective maximum and minimum distances, with $D(s_{j},BS)$ representing the distance from node *j* to the base station.

Regarding the fitness function:(22)$$f=\omega \times f_{1}+\left(1-\omega \right)\times f_{2}.$$
*w* is the weight factor.

### 4.4. Epoa-Chs Pseudocode

In each iteration of the particle swarm, all particles evaluate their fitness based on their current position (solution), compare it with their own best-known position (local best) and the current best position in the swarm (global best). If the fitness is improved, the individual and/or swarm best values are updated. After each iteration, the current best swarm position is compared with the previous best swarm position. If the fitness has improved, the swarm’s best position is updated. This process continues until the fitness requirements are met or the iteration is completed, at which point the best candidate solution is output.The pseudo-code of EPOA-CHS is shown in Algorithm 1.
**Algorithm 1** EPOA pseudo-code**Require:** Cluster heads of the HWSN**Ensure:** Optimal cluster head Step 1: Initialize the population of Pelican as a matrix X according Logistic-Sine Chaotic map using Equations (11)–(14); Step 2: Calculates the fitness values for each row in X using Equations (20) and (21); Step 3: **while** 
$t<t_{max}$
** do** Update the best condidate solution $X_{best}^{t}$ and Get the fitness best value according minimum of the fitness; Update the location of the $X_{prey}^{t}$; Calculate the fitness f($X_{prey}^{t}$);   **for** i=1:N **do**     /*phase 1:Moving towards prey(exploration phase)*/     Update the location of the $X_{new}^{t}$ using Equation (Equation 4);     Calculate the fitness $f(X_{new}^{t}$);     **if** $f(X_{new}^{t}$) <f($X_{i}^{t}$) **then**        $X_{i}^{t}\leftarrow X_{new}^{t}$;        $f\left(X_{i}^{t}\right)\leftarrow f\left(X_{new}^{t}\right)$;     **end if**     /*end phase 1*/     /*phase 2:Winging on the water surface (exploitation phase)*/     Update the location of the $X_{new}^{t}$ using Equations (4), (5), and (15)–(19);     Calculate the fitness $f(X_{new}^{t}$);     **if** f($X_{new}^{t}$) < f($X_{i}^{t}$) **then**        $X_{i}^{t}\leftarrow X_{new}^{t}$;        $f\left(X_{i}^{t}\right)\leftarrow f\left(X_{new}^{t}\right)$;     **end if**     /*end phase 2*/   **end for**   Save best condidate solution $X_{best}^{t}$ and the fitness best value;   t = t + 1**end while**Step 4: Repeat Step 3 until it reaches the maximum number of iterations;Return $X_{best}^{t}$ /*Optimal cluster head*/

## 5. Results and Discussion

To evaluate the performance of the EPOA-CHS algorithm, we performed simulations using the MATLAB software. To ensure a fair comparison, we conducted the evaluation under identical experimental conditions, including the SEP, DEEC, Z-SEP, and PSO-ECSM algorithms. By comparing the results obtained from these various algorithms, we can assess the efficacy and superiority of the EPOA-CHS algorithm in terms of energy efficiency and other relevant performance metrics.

### 5.1. Simulation Settings

This technique uses a flat network model with nodes distributed at random across a 100 × 100 m${}^{2}$ region. The suggested approaches are contrasted with the following algorithms: SEP, DEEC, Z-SEP, and PSO-ECSM. All algorithms use the same set of input setup settings for the network. In addition, the base station (BS) is positioned in the network’s upper-center, guaranteeing that every sensor node has at least one neighbor. Each sensor node has the ability to send data packets in a single hop to the BS. Furthermore, the sensor nodes exhibit heterogeneity and possess the same communication range. Further details regarding the system can be found in Table 1.

### 5.2. Residual Energy

The technology’s advanced CH selection method leads to better performance and effectively achieves higher average residual energy. This leads to minimized premature node failures and an extended network lifetime. The EPOA-CHS algorithm demonstrates a higher average residual energy, indicating enhanced network survivability. Additionally, the smaller energy deviation observed in EPOA-CHS signifies its ability to effectively reduce energy imbalances, preventing premature failures caused by excessive energy consumption.

As a result, the energy consumption among nodes using EPOA-CHS is more evenly distributed, leading to a more balanced network operation. Combined with the additional energy savings achieved by the algorithm, this results in an increased network lifetime compared to the other four algorithms. Figure 2a–d illustrate the network’s total residual energy. It is obvious that EPOA-CHS exhibits higher total residual energy, indicating an extended network lifespan.

### 5.3. Network Lifetime

By considering various proportions of advanced nodes in the simulation, Figure 3 shows the advantages of the EPOA-CHS algorithm in terms of network lifetime. The EPOA-CHS method significantly extends the network lifespan in Figure 3a where the energy multiplier for advanced nodes is α = 0 and the proportion of advanced nodes is m = 0. Specifically, it shows a relative increase of 246.4%, 214.2%, 389.0%, and 65.0% compared to the SEP, DEEC, Z-SEP, and PSO-ECSM algorithms, respectively. Moving to Figure 3b, with m = 0.1 and α = 2, the EPOA-CHS algorithm still outperforms the other protocols with an increase of 100.1%, 116.8%, 91.2%, and 22.4% in network lifespan compared to SEP, DEEC, Z-SEP, and PSO-ECSM, respectively. Similarly, in Figure 3c (m = 0.2 and α = 3) and Figure 3d (m = 0.3 and α = 1.5), the EPOA-CHS algorithm maintains its superiority, exhibiting improvements of 73.7%, 131.4%, 103.3%, and 21.5% in network lifespan over SEP, DEEC, Z-SEP, and PSO-ECSM for m = 0.2, and 70.2%, 120.2%, 91.2%, and 21.2% for m = 0.3, respectively. The results clearly demonstrate that the EPOA-CHS algorithm surpasses the tested protocols in terms of efficacy and network lifespan extension.

Table 2 and Figure 4 show the simulation results for the network life cycle at various percentages of advanced nodes for the first node to die (FND, First Node Death), 50% of nodes to die (HND, Half Nodes Death), 70% of nodes to die (MND, Most Nodes Death), and all nodes to die (LND, Last Node Death). And as can be seen from these two tables, we perform better on the time of death for other protocols. And the table also clearly shows that the EPOA-CHS algorithm, compared with other protocols, is also dominant in the unstable time of the network. This is because EPA-CHS takes into account the heterogeneity and global optimization ability, which makes the distribution of node energy use more balanced and reduces the unstable period of the network.

### 5.4. Alive Nodes Number

Figure 5a–d illustrate the number of active nodes in each cycle of a network comprising 100 nodes. In this context, the term “active nodes” refers to the count of live nodes, i.e., nodes that still have remaining energy. Conversely, the number of death nodes corresponds to the count of nodes that have depleted their energy and are no longer operational.

The increased number of living nodes for the EPOA-CHS algorithm when compared to the other four algorithms demonstrates its better energy consumption performance. This suggests that the nodes’ energy consumption is distributed more evenly. The number of surviving nodes is a crucial factor, but it is not the only one that determines the network’s lifespan. Data transmission and network connectivity depend on the existence of a feasible path linking the remaining nodes to the BS.

For the network to remain connected, it is crucial to have a sufficient number of surviving nodes, but the effectiveness and balance of energy consumption are equally important. The EPOA-CHS technique effectively maintains network connectivity and achieves energy balance, contributing to the network’s extended lifetime.

### 5.5. Packet Delivery

Figure 6a–d show how many data packets the BS has received, giving information on network throughput. In Figure 6a, with m = 0 and α = 0, the EPOA-CHS algorithm demonstrates improved network throughput compared to SEP, DEEC, Z-SEP, and PSO-ECSM. The network throughput of EPOA-CHS specifically improved by 102.3%, 94.8%, 69.8%, and 8.67% in comparison to SEP, DEEC, Z-SEP, and PSO-ECSM, respectively. In Figure 6b, where m = 0.1 and α = 2, the EPOA-CHS algorithm continues to exhibit superior network throughput, with an increase of 87.9%, 79.4%, 36.6%, and 6.86% compared to SEP, DEEC, Z-SEP, and PSO-ECSM, respectively. Figure 6c represents the results for m = 0.2 and α = 3, showing that the network throughput of EPOA-CHS improved by 56.8%, 53.2%, 25.4%, and 5.61% relative to SEP, DEEC, Z-SEP, and PSO-ECSM. Finally, Figure 6d demonstrates the performance for m = 0.3 and α = 1.5, with the EPOA-CHS algorithm achieving network throughput increases of 87.6%, 78.5%, 33.7%, and 4.76% compared to SEP, DEEC, Z-SEP, and PSO-ECSM, respectively. These results clearly highlight the enhanced network throughput achieved by the EPOA-CHS algorithm across different scenarios and its superiority over the evaluated protocols.

## 6. Conclusions

In this research, we have presented a cutting-edge method for energy-constrained wireless sensor networks (WSNs) dubbed EPOA-CHS. The EPOA-CHS technique incorporates the use of the logistic-sine chaotic map for population initialization, providing enhanced randomness and diversity in the initial population. The aim of the EPOA-CHS method is to select cluster heads (CHs) based on fitness functions that take into account various sensor properties. Additionally, the EPOA-CHS technique integrates the traditional EPOA algorithm with the Levy function during the winging phase, enabling improved exploration and convergence capabilities. We assessed the EPOA-CHS technique’s performance using a series of simulations and contrasted it with current methods.The outcomes show how effective the EPOA-CHS method is in terms of energy use and overall network performance. Moving forward, our future research aims to extend the application of the EPOA-CHS technique to the multihop routing process in large-scale WSNs. We foresee additional advancements in energy efficiency and scalability by resolving the difficulties involved with routing in such networks.

## Figures and Tables

**Figure 1 sensors-23-07711-f001:**
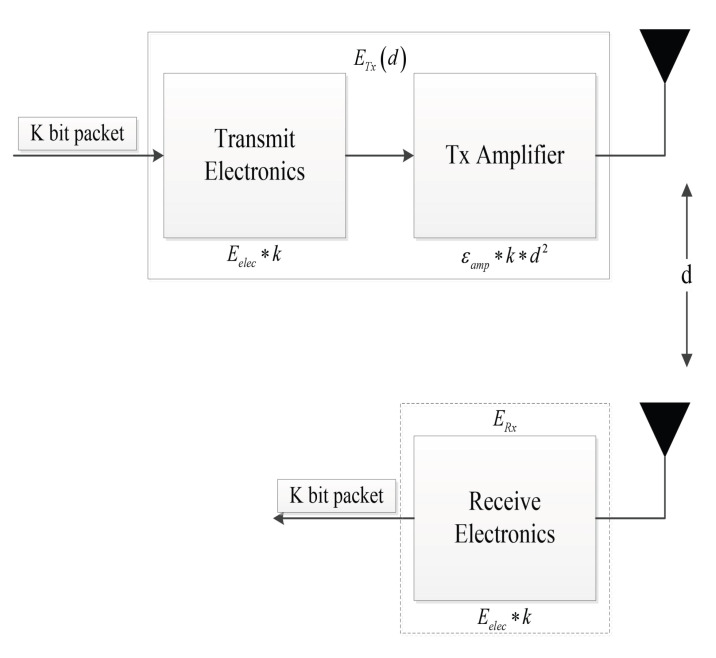
First-order radio energy model.

**Figure 2 sensors-23-07711-f002:**
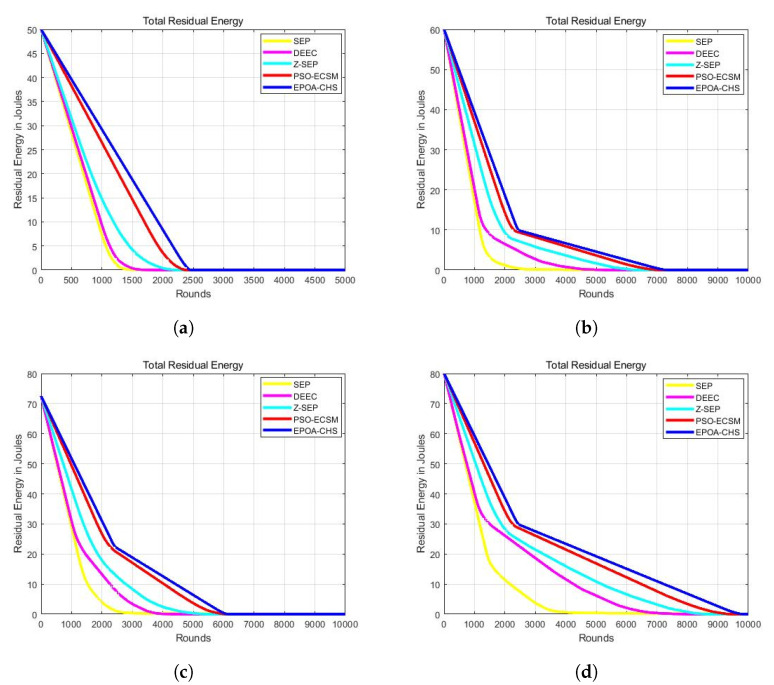
(**a**) m = 0, α = 0, (**b**) m = 0.1, α = 2, (**c**) m = 0.2, α = 3, (**d**) m = 0.3, α = 1.5.

**Figure 3 sensors-23-07711-f003:**
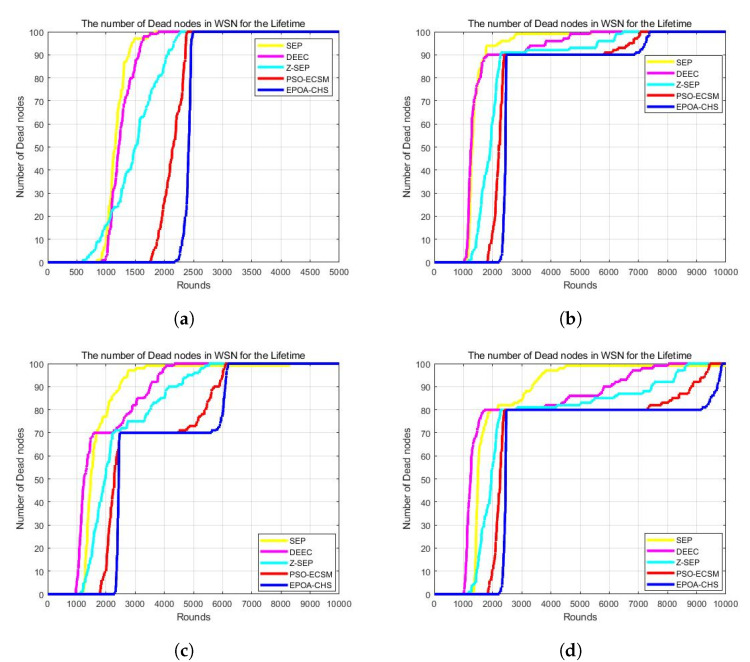
(**a**) m = 0, α = 0, (**b**) m = 0.1, α = 2, (**c**) m = 0.2, α = 3, (**d**) m = 0.3, α = 1.5.

**Figure 4 sensors-23-07711-f004:**
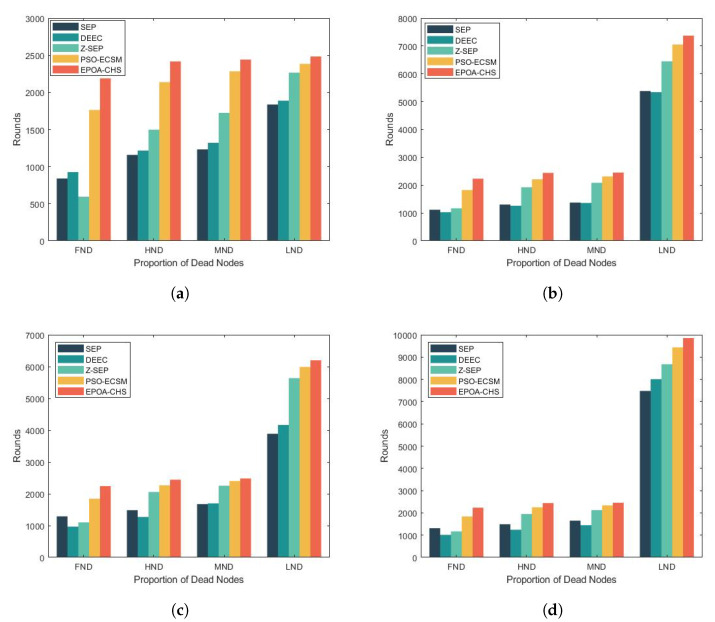
(**a**) m = 0, α = 0, (**b**) m = 0.1, α = 2, (**c**) m = 0.2, α = 3, (**d**) m = 0.3, α = 1.5.

**Figure 5 sensors-23-07711-f005:**
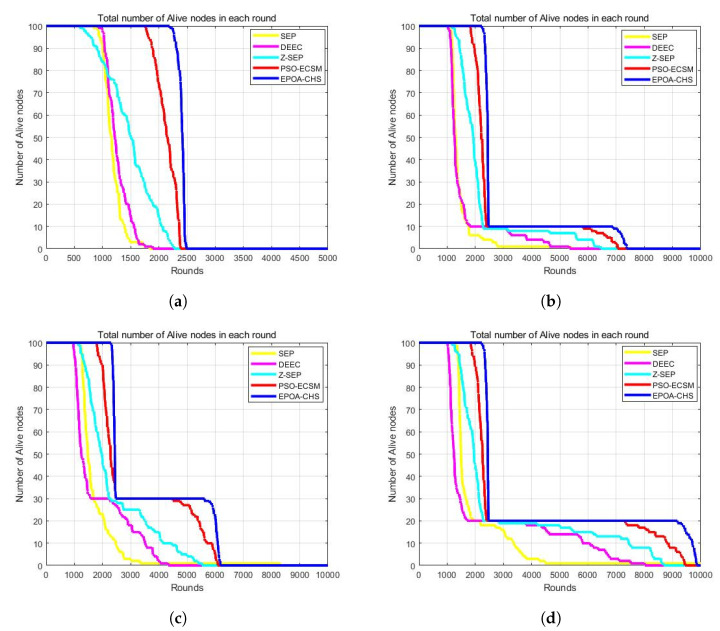
(**a**) m = 0, α = 0, (**b**) m = 0.1, α = 2, (**c**) m = 0.2, α = 3, (**d**) m = 0.3, α = 1.5.

**Figure 6 sensors-23-07711-f006:**
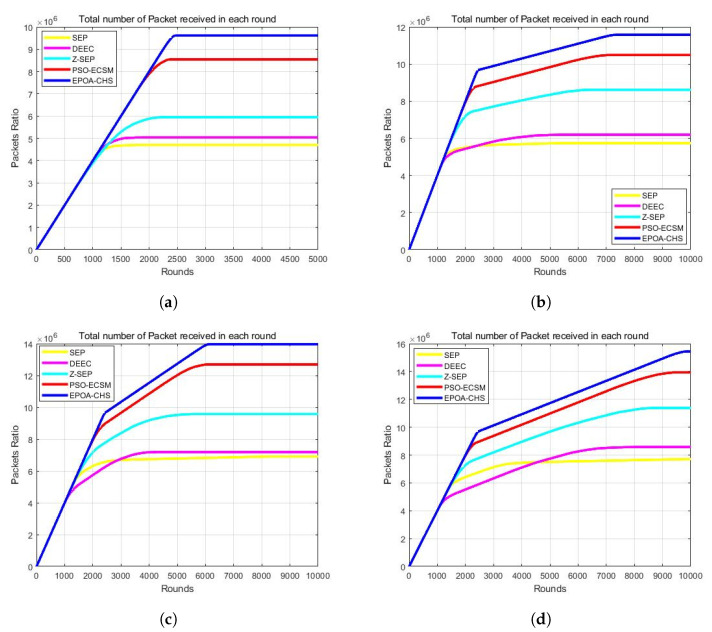
(**a**) m = 0, α = 0, (**b**) m = 0.1, α = 2, (**c**) m = 0.2, α = 3, (**d**) m = 0.3, α = 1.5.

**Table 1 sensors-23-07711-t001:** Simulation correlation parameter.

Parameters	Value
Network Field	(100,100)
Number of nodes	100
$E_{0}$	0.5 J
Packet Size	4000 Bits
$E_{elec}$	50 nJ/bit
$E_{fs}$	10 nJ/bit/m${}^{2}$
$E_{amp}$	0.0013 pJ/bit/m${}^{4}$
$E_{DA}$	5 nJ/bit/signal
$d_{0}$	70 m
$P_{opt}$	0.1
Fraction of the	m = 0, m = 0.1,
advanced nodes	m = 0.2, m = 0.3
Times more energy	α=0,α=2,
than normal nodes	α=3,α=1.5

**Table 2 sensors-23-07711-t002:** Number of rounds and network life cycle.

No. of Rounds					
**Cases for Heterogeneity**	**Protocol**	**FND**	**HND**	**MND**	**LND**
	SEP	840	1158	1232	1837
	DEEC	926	1216	1321	1888
	Z-SEP	595	1498	1724	2266
m = 0, ***α* = 0**	PSO-ECSM	1763	2139	2285	2384
	EPOA-CHS	2910	2417	2442	2484
	SEP	1116	1302	1372	5383
	DEEC	1030	1259	1361	5341
	Z-SEP	1168	1923	2086	6450
	PSO-ECSM	1825	2212	2316	7055
m = 0.1, ***α* = 2**	EPOA-CHS	2233	2441	2452	7373
	SEP	1291	1486	1676	3888
	DEEC	969	1274	1699	4165
	Z-SEP	1103	2055	2256	5638
	PSO-ECSM	1846	2269	2402	5989
m = 0.2, ***α* = 3**	EPOA-CHS	2243	2445	2482	6198
	SEP	1312	1491	1652	7480
	DEEC	1014	1243	1448	8088
	Z-SEP	1168	1951	2125	8677
m = 0.3, ***α* = 1.5**	PSO-ECSM	1842	2253	2340	9436
	EPOA-CHS	2233	2443	2458	9856

## Data Availability

Not applicable.

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
