# Peer review of "Enhanced Pelican Optimization Algorithm for Cluster Head Selection in Heterogeneous Wireless Sensor Networks"

_sensors, 2023, doi:10.3390/s23187711_

Round 1

Reviewer 1 Report

1. Affiliation of all authors are missing

2. Abstract should be improved, particularly on how the proposed algorithm can achieve efficient energy saving through optimal clustering. The challenges of existing clustering methods should be briefly explained too.

3. Quantitative results need to be presented in Abstract.

4. Problem statement and research gaps that lead to current work are not well explained. It is not clear what are the challenges encountered in existing studies that lead to the proposal of current works. It is also not mentioned why there is a need of defining new fitness function? Further classification are needed from authors to justify the significance of current work.

5. The advantages of using swarm intelligence and metaheuristic methods to solve the CH selection problem in HWSNs should be explained.

6. The forth contribution does not look appropriate and needs to be revised. 

7. It is necessary for authors to further explain the main differences between the proposed work and current work. 

8. Some of the abbreviations such as SEP, DEEC and etc are used without defining their full terminology first.

9. What do the author means by "diversity of possible solution" in Line 135? I don't think the population size is directly related to solution diversity.

10. Eqs. (2), (3), (5) and other similar equations are not properly presented. The if-else statement needs to be specified clearly. It is also not necessary to present the equations in this bracket {}. 

11. Please check the index i in Line 151 refers to dimension index. It seems to be a solution index to me.

12. Connection between Eq. (3) and (4) are not observed. What happen to X_i_Pi obtained in (3) and how it is used in Eq. (4)?

13. The symbols used to define number of iterations measured in Lines 168 and 169 are not defined.

14. What do the author mean by "Where Mrepresents the number of nodes and is equal to Ntimes M." in Line 181? This statement is confusing. 

15. Some symbols inside Fig. 1 cannot be seen clearly. Please enlarge the symbols to improve the clarity of figure.

16. The search mechanisms of EPOA in Section 4.2 needs more detailed explanation. Currently, the authors only introduce the modifications made but they did not clearly explain how these modifications are incorporated into the original POA? 

17. Furthermore, it is necessary for the authors to justify how the modifications made can address the fundamental issues of metaheuristic methods such as POA, particularly on the balancing of exploration and exploitation searches.  

18. Are there any specific reasons to choose logistic map, sine map, and logistic-sine map for population initialization? Why 3 chaotic maps are used instead of other numbers?

19. Please show the solution encoding process. It is not clear what decision variables to be optimized in this study and their boundary limits.

20. What are the parameter w shown in Line 283? How to determine its value? Moreover, an equation numbering should be provided to this equation. 

21. The psuedocode of EPOA presented is too brief. Some important steps such as fitness evaluation is not observed during the iterative search process. Significant improvement is needed.

22. What is the actual symbol used to represent the dimensional size of problem? Sometimes it is represented as m, sometimes as D. Please standardize it.

23. Some descriptions should be provided to explain the algorithms selected for comparison, i.e., SEP, DEEC, Z-SEP and PSO-ECSM. There are no references provided for these selected algorithms for readers to find out more of their information. 

24. Convergence curves and other graphs presented in Figs. 2,3, 4, 5 and etcs are too small and needs to be enlarged. 

Some grammatical errors and typos can be found. Authors need to proofread the manuscript again before resubmission.

Author Response

Thank you very much for your comments. I apologize for the inconvenience caused to your review due to some problems in the paper. We have proofread the manuscript and the documents submitted below have been revised according to your suggestions.

Reviewer 2 Report

1. Give the whole words only when the abbreviation first appears. WSN, CH and so on are given again and again.

2. This work utilize Levy function to improve PSO algorithm. Why takes one section to introduce Pelican Optimization Algorithm?

3. The author didn't verify the optimal and convergence of the improved PSO. And then, the performance of this algorithm is not sure.

4.The compared algorithms are not proper. Some are too old, such as SEP and  DEEC. Besides, except PSO, there are some heuristic algorithms are used, such as GA. Please provide more comparisons.

5. Despite reducing energy by improving clustering, there are some works reduce dead nodes through combing clustering and node scheduling, such as "an energy efficient routing protocol for 3D wireless sensor network". Please compare with this kind of work.

there are some non-standard academic expression, which should be corrected, such as the first sentence of the last paragraph in section 1. 

some grammar errors exsit, such as the last paragraph in subsection 4.3.

Author Response

Thank you very much for your suggestion. I apologize for the inconvenience caused to your review due to some problems in the paper. We have proofread the manuscript and the documents submitted below have been revised according to your suggestions.

Reviewer 3 Report

Minor revision will be needed for further processing.

Author Response

We would like to thank the reviewers for volunteering the time to review our manuscript .We have made further revisions to the full text.

Round 2

Reviewer 1 Report

Authors have addressed most of my comments raised in previous review cycle. Just one minor issue with previous Point 10. It seems that the authors have misunderstood my meaning. "{" is still needed in some equations such as Eq. (2), (3), (5) and etc since there are more than one condition are mentioned in these equations. Only "}" needs to be excluded.

Author Response

Thank you very much for your suggestions. We have revised the paper according to your suggestions.

Reviewer 2 Report

Almost my questions have not been well addressed, such as, for the first question, just the full words and the abbreviation of CH appear in Columns 34, 143, 206, 284, etc.

Besides, there are still some expression problem. Here are just some examples.

Section lacks the necessary description of Algorithm 1.

The title of the subfigure is too verbose. For example, Figure 2(a) can just use m=0, \alpha= 0.

A space is need between a word and a punctuation.

The mathematical notation is de-normalized, such as N which is in the highlighted sentence in Col. 163.

More explanation of results should be included, especially the reason of inflection points.

It's ok.

Author Response

(The authors gave the same response as above.)
